# How to Identify Future Priority Areas for Urban Development: An Approach of Urban Construction Land Suitability in Ecological Sensitive Areas

**DOI:** 10.3390/ijerph18084252

**Published:** 2021-04-16

**Authors:** Xiaobo Liu, Yukuan Wang, Ming Li

**Affiliations:** 1Institute of Mountain Hazards and Environment, Chinese Academy of Sciences, Chengdu 610041, China; xbliu@imde.ac.cn (X.L.); liming@imde.ac.cn (M.L.); 2University of Chinese Academy of Sciences, Beijing 100049, China; 3College of Geography and Resources Science, Neijiang Normal University, Neijiang 641100, China

**Keywords:** land suitability, spatial pattern, the upper reaches of the Yangtze river, mountain area

## Abstract

The suitability of urban construction land (SUCL) is key to the appropriate utilization of land resources and represents an important foundation for regional exploration and land management. This study explores the SUCL conceptual framework by considering the theory of human-land relationships. The upper reaches of the Yangtze River were studied, a typical ecologically-sensitive area of China. The spatial pattern and control of the SUCL were determined using the improved entropy method. The results show that an area of 91 × 10^4^ km^2^ was categorized as prohibited or restricted, and these categories account for 28.61% and 50.66% of the total area, respectively. Priority areas and suitable areas are mainly located in the Chengdu Plain, the urban agglomeration of southern Sichuan Province, Chongqing, and the economic corridor in the west, and the surrounding cities of Guiyang and Kunming. SUCL hotspots feature obvious spatial heterogeneity and are concentrated in Sichuan Basin and Guizhou Plateau. The SUCL is obviously constrained by the physical geography of this region. In addition, towns affected by the pole–axis effect have stronger suitability for development and construction. These findings will be very useful for land managers as they provide relevant information about urban development in mountainous areas.

## 1. Introduction

For the past 40 years, in China, urban construction land has expanded at an unprecedented speed in response to the booming economy [1,2]. The annual growth rate of urban construction land in China is 4%, which far exceeds the average 1.2% of developed regions across the world [3]. Moreover, urbanization has increased from 17.9% in 1978 to 60.6% in 2020. This rapid expansion of urban construction land will still remain the main trend for Chinese urban land use over the coming decades [4,5]. However, urban land resources have been overused for a long time, and are facing enormous pressure for further development [6]. In fact, the shortage of land resources, characterized by an accelerated consumption of urban construction land resources, has become the main bottleneck affecting the healthy and sustainable development of Chinese cities [1,7]. Rapid urban expansion not only causes impacts threatening the ecological environment and the protection of important farmland, it also leads to the loosening of the urban structure without management; this decreases the spatial efficiency of a city and hinders its healthy development [8]. Over the next 30 years, China’s industrialization, urbanization, and population will successively continue to peak, which will drive a further demand for urban construction land [9]. Moreover, China’s large population requires the retaining of adequate farmland to guarantee grain supply for more than 1.4 billion persons [10,11]. It can be expected that both the supply and demand of urban construction land in Chinese cities will become more severe in the future. To maintain an effective balance between grain yield and urbanization, China needs to fully use every inch of urban construction land. Hence, the priority question is how to fully utilize these precious land resources in the process of urbanization while minimizing negative effects with the goal to achieve sustainable development of the urban ecosystem. In this regard, land suitability evaluation plays an important role [12,13].

Land suitability represents whether a particular type of land is suitable to support a defined land use, either in its current state or after specific improvements. Land suitability is the premise of land use planning because it represents the strongest advantages of different land use types. The chief aim of land suitability evaluation is the prediction of the future land performance based on its attributes, which helps to improve the land use efficiency and obtain the optimal benefit from the land [14]. This process includes the selection, identification, and description of the types of land use relevant to the area under consideration [15], the mapping and description of the types of land that appear in the area, and the evaluation of the suitability of each type of land for the selected land use types. Suitability evaluation of urban construction land (SEUCL) is a type of land suitability evaluation. Its fundamental task is the depiction of the spatial heterogeneity of land [16], to obtain core data of urban land management and land use planning. In addition, SEUCL can recognize the spatial difference of regional suitable urban construction land, determine the reasonable extension direction of construction land, and identify new urban land space [17]. At the same time, it can decrease the loss of life and property caused by natural disasters such as debris flows and landslides, and provide a safe and healthy living environment for residents.

Over the last ~20 years, the techniques of SEUCL have increasingly become integral components of urban, environmental, and regional planning activities [18]. With regard to evaluation models, in general, the existing evaluation models for SEUCL can be classified into three categories: The first category uses early hand-drawn overlay techniques. With regard to this approach, the work of McHarg is generally considered to be of pioneering significance [19]. The second category is the multi-criteria evaluation based on GIS, which contains a series of methods, such as weighed ideal point method, linear combination (WLC) [20], ordered weighted averaging (OWA) [21,22], and concordance analysis [23]. The third category is a recently developed approach that focuses on artificial intelligence technologies, which includes artificial neural networks, genetic algorithms, fuzzy logic techniques, and cellular automata. The most popular method is the use of overlay mapping as a framework to evaluate land suitability in conjunction with statistical methods [24,25]. However, the above-mentioned evaluation models are subject to following limitations: (1) The existing research mainly considers the improvement of evaluation methods, while ignoring the spatial clustering and differentiation characteristics of urban land suitability. Such an approach is not conducive to the rational formulation of large-scale urban construction land space planning. (2) Almost all evaluation models need to select evaluation indexes and assign appropriate weights [26,27]. With regard to index selection, different evaluation indexes should be selected to evaluate different regions, which has become the consensus in academic circles [28,29,30]. Although differences exist with regard to research focuses or methods, and no unified index selection criteria have been formulated to date, there seems to be a common trend in the selection of evaluation indexes for the suitability of urban construction land (SUCL) [31]. Relevant literature tends to analyze problems from the perspectives of development and protection to promote the effective use of land resources, rather than stressing to focus on the coordination of multi-factors and the holistic evaluation approaches of human–Earth interaction. Moreover, previous studies mainly emphasized the threshold identification of a single study area based on relevant indicators, such as the area of a city or its population [32]. However, such approaches lacked the necessary comparison of the inter-regional urban development potential based on indicators with increased information richness. Furthermore, in the design of the evaluation system, most previous works focused on simple superpositions of multiple statistical indicators, while lacking in-depth mechanistic explorations. Therefore, it is a very meaningful and challenging task to modify the existing index system and increase its suitability for ecologically fragile mountain areas. The determination of the index weight is another important step of the evaluation process, and the weight assigned to each index is one of the most sensitive parameters in SEUCL and also a potential source of considerable uncertainty [33]. For example, the analytical hierarchy process (AHP) is one of the most popular methods for calculating criteria weights in SEUCL via an expert pair-wise comparison matrix [23]. Using their weights, criteria can be subsequently aggregated into a single imprecise SEUCL estimation point, which results in uncertainties without confidence. Therefore, the entropy method was improved to eliminate the 0 value from the SEUCL calculation and improve both the accuracy and confidence of weight allocation.

China has the widest distribution of mountainous regions in the world, and mountains, plateaus, and hills account for ~65% of the Chinese territory [34]. Mountainous areas account for 94.7% of the upper reaches of the Yangtze River [35], and represent a very fragile ecosystem that is seriously threatened by increasingly intensive and unsustainable urban development and construction [36]. In 2016, the construction of the Yangtze River Economic Belt became a national strategy, and consequently, the development and protection of urban land in the upper reaches of the Yangtze River attracted significant attention. Reducing the negative effects generated in the evolution of human-land relationships is a practical problem that needs to be urgently solved. This paper evaluates the SUCL at the county level in the upper reaches of the Yangtze River from the perspective of the human–Earth relationship. Furthermore, this paper investigates the spatial distribution characteristics of the SUCL in the upper reaches of the Yangtze River. The specific goals are summarized in the following: (1) To establish a multidimensional conceptual framework for the SEUCL of the coupled human-land system, and then apply it to evaluate the SUCL; (2) to explore the spatial distribution characteristics and mechanisms of the SUCL, and provide a basis for future urban construction layout.

## 2. Materials and Methods

### 2.1. Study Area

The upper reaches of the Yangtze River are located at 24°50′–35°35′ N and 90°30′–112°04′ E in the southwestern part of China. The watershed covers a region from the origin of the river to Yichang of Hubei Province, a total length of 4511 km [37]. The total area is about 115 × 10^4^ km^2^ with a total population exceeding 163 million at the end of 2018, which accounts for 58.9% of the area and 35.4% of the population in the whole Yangtze valley, respectively [38]. The tributaries include the Yalong River, the Minjiang River, the Jialing River, and the Wujiang River. The total cultivated land of the upper reaches of the Yangtze River is 21.17 × 10^4^ km^2^; the forest and grassland are 81.40 × 10^4^ km^2^; the construction land is 1.01 × 10^4^ km^2^; and the urban construction land is 0.33 × 10^4^ km^2^, accounting for 19.27%, 70.78%, 0.88%, and 0.29% of the total area, respectively. At present, the economic development of the upper reaches of the Yangtze River lags far behind that of the middle and lower reaches of the Yangtze River, with a small proportion of the total economic output of China and a per capita GDP that is far below the national average [39]. Achieving a balanced regional development through urban construction and economic development is an urgent task. At the same time, more than 90% of the land in the upper reaches of the Yangtze River consists of mountainous areas and plateaus, with a very fragile and sensitive ecosystem [37]. If the urban construction land is not allocated reasonably, problems of soil erosion, ecological damage, and natural disasters are easily aggravated. These realities imply that balancing urban development and ecological protection is a complex and challenging task. The upper reaches of the Yangtze River are a physical geographical concept. For the convenience of statistical analysis and the classification of evaluation results, this paper uses county-level administrative areas flowing through the Yangtze River Basin as the basic unit of analysis. Based on this principles, the research scope covers 346 county level administrative regions, in which Sichuan Province consists of 182 regions, Guizhou Province consists of 45 regions, Yunnan Province consists of 44 regions, Chongqing City consists of 37 regions, Hubei Province consists of 9 regions, Qinghai Province consists of nine regions, Shanxi Province consists of four regions, and Tibet Autonomous Region consists of three regions (Figure 1).

### 2.2. Data Sources and Processing

The data used in this study were derived from a variety of sources. Data can be divided into two types: Governmental statistics data and spatial data. Governmental statistics include the GDP, the permanent population (for 2015 and 2019), the investment in fixed assets, the urban population, the proportion of the tertiary industry, the total retail sales of consumer goods, and public infrastructure spending. These data are obtained from the statistical yearbooks of the provinces, prefecture-level city statistical yearbooks, and county-level statistical bulletins of national economic and social development of the year 2019. The spatial data mainly includes land-use distribution data at a resolution of 30 × 30 m, water resource distribution vector data, geological disaster data, and high-precision digital elevation model (DEM) data at a resolution of 30 × 30 m. This dataset was provided by the national Earth system science data center (http://www.geodata.cn (accessed on 6 September 2020)). Other data, such as city and county administrative maps, provincial road networks, as well as freeway, railway, and national highways, were obtained from the National Basic Information Center (http://ngcc.sbsm.gov.cn (accessed on 24 October 2020)).

### 2.3. Methods

#### 2.3.1. Conceptual Framework for SEUCL

Important tasks faced by studying SEUCL include the establishment of common conceptual frameworks and research paradigms based on the theory of the human–Earth relationship. As shown in Figure 2, the giant human–Earth relationship system consists of two subsystems: The land ecosystem and the human activities system. Land development is a process in which humans directly affect the surface environment of the Earth, and the result is the efficiency of land use [40]. Land provides the spatial support for human survival and development, and humans transform and utilize the land to achieve sustainable development. Humans have had a dual role of both negative and positive interveners [41]. On the one hand, strong human activities and excessive land utilization may destroy the initial state of land ecosystems to some degree [42]. On the other hand, certain policies and technologies may promote the benign development of land ecosystems [43].

Therefore, SUCL, embodying the human–Earth relationship, can be decomposed into two parts as follows: (1) Carrying capacity of the land ecosystem. The global impact of an increasing population, combined with resource and environmental constraints, highlights the urgent need for humanity to live within the carrying capacity of the available land [44,45]. Such a carrying capacity is composed of resources and the environment [46]. The resources carrying capacity reflects the supply of resources needed by urban areas, such as land resources suitable for urban construction and water resources required for the survival of the urban population. The carrying capacity of the ecological environment actually reflects the constraints the natural environment imposes on urban construction expansion, which are most obvious in ecologically vulnerable regions or ecologically sensitive regions. Suitability evaluation prevents ecological imbalance or environmental pollution caused by urban construction [47]. However, little attention has been directed to quantifying this land carrying capacity, nor to accurately determining where cities should be placed within a particular area of land. Therefore, part I of SUCL is generated: Resource environmental carrying capacity (i.e., ecological environment cognition). (2) Construction suitability evaluation. In primitive societies, human demand for land space secures subsistence food through single production labor. With the expansion of the scale and type of human needs for survival and development, both the utilization and transformation of the original land have been continuously deepened [48]. New land use patterns are continuously forming, ultimately resulting in differences between urban lands and other types of lands. Different from agricultural lands, a city is the product of concentration of human activities and it is a region with the most concentrated and complicated land functions. A city is also the product of continuous superposition of human demands on land functions. Nevertheless, not all lands in all regions are suitable for urban construction. Regions with relative population concentration, dense economic investments, and developed traffic networks are often ideal places for future urban construction. This represents the territory function of urban construction. Hence, it is necessary to determine whether land functions can support urban construction to determine regions for future urban development. This is Part II of urban land suitability evaluation: Construction suitability evaluation.

The above framework clearly shows that resource environmental carrying capacity evaluation and construction suitability evaluation are two basic elements of SEUCL, which form a parallel relationship. The resource environmental carrying capacity reflects the basic supply conditions to support urban construction. Construction suitability reflects the potential or possibility of the land to support future urban constructions rather than the possibility of lands to be used for other purposes (e.g., cultivated land and forest land).

#### 2.3.2. Multi-Factor Comprehensive Evaluation Model

The multi-factor comprehensive evaluation model is derived from the simple-factor evaluation model, and its improvement can be found in the use of weighted sum or Boolean operations. On the basis of single-factor evaluation at the feature level, the weighted sum method is used to merge the integral step by step [49]. Spatial vector maps, such as land use, elevation heights, slope, and geological hazard, were drawn in GIS as input multi-factor layers, to evaluate SUCL in the study area [1]. Similar models, based on multi-factor evaluation and geospatial technologies, have proven to be efficient, valuable, and technologically sound means to analyze land suitability, as well as to monitor urban growth [3]. The natural conditions of the upper reaches of the Yangtze River are complex, and the economic development level gap is large. Therefore, a multi-factor comprehensive evaluation model can better reflect the regional differentiation characteristics of the physical geographical environment as well as both social and economic factors in the area. This improves the reliability of the evaluation results. The calculation formula is:(1)$$S=\sum_{i=1}^{n}w_{i}x_{i}$$ where S represents the total score of evaluation unit; $w_{i}$ represents the weight value of subfactor i; and $x_{i}$ represents the score of subfactor i.


#### 2.3.3. Hot Spot Analysis (Getis-Ord Gi*)

Given a set of weighted features, hot spot analysis tool (Getis-Ord Gi*) uses the Getis-Ord Gi* statistic to identify statistically significant hot spots and cold spots. The Getis-Ord Gi* statistic is then calculated for each feature in a dataset [45,50]. The resultant z-scores and p-values can identify where features with either low or high values cluster spatially.

The Getis-Ord local statistic is given as:(2)$$G_{i}^{*}=\frac{\sum_{j=1}^{n}w_{i,j}x_{j}-\bar{X}\sum_{j=1}^{n}w_{i,j}}{S\sqrt{\frac{\left[n\sum_{j=1}^{n}w_{i,j}^{2}-\left(\sum_{j=1}^{n}w_{i,j}\right)\right]}{n-1}}}$$
where $x_{j}$ represents the attribute value for feature j; $w_{i,j}$ represents the spatial weight between feature i and j; and n is equal to the total number of features:(3)$$\bar{X}=\frac{\sum_{j=1}^{n}x_{j}}{n}$$
(4)$$S=\sqrt{\frac{\sum_{j=1}^{n}x_{j}^{2}}{n}-{\left(\bar{X}\right)}^{2}}$$

The statistic is a z-score; therefore, not further calculations are required.

#### 2.3.4. Evaluation Index Weights

This study uses the improved entropy method to calculate the weight of the suitability evaluation of urban construction land index system. The entropy method is an objective method to determine weights. Entropy is a measure of disorder within a system and can be used to measure the total amount of information and the weight of known data [51]. This method is widely used for comprehensive evaluations [52].

(1) Dimensionless processing

When the index weight is calculated using the entropy method, the data should be subject to dimensionless processing. Here, the standardized translation method is used to perform data dimensionless processing. The formula is as follows:(5)$$x_{ij}^{\prime }=\{{}_{(x_{j\max}-x_{ij})/(x_{j}{}_{\max}-x_{j\min})\mathrm{Negative}\;\mathrm{indicators}}^{(x_{ij}-x_{j\min})/(x_{j}{}_{\max}-x_{j\min})\mathrm{Positive}\;\mathrm{indicators}}$$ where $x_{ij}^{\prime }$ represents the value of dimensionless processing of item j, and $x_{ij}$ represents the value of the jth indicator of the ith object (i = 1, 2, …, n, j = 1, 2, …, m). $x_{j\min}$ represents the minimum value of item j, and $x_{j}{}_{\max}$ represents the maximum value of item j.

(2) Translation coordinates (improve)

To eliminate the 0 value of $x_{ij}^{\prime }$, an appropriate positive number 2 can be selected to translate the coordinates.
(6)$$x^{+}=x^{\prime }{}_{ij}+2$$

(3) Calculate the entropy of the jth indicator
(7)$$H_{i}=-k\sum_{j=1}^{n}x_{ij}^{+}\ln x_{ij}^{+},(i=1,2,3\cdots m;j=1,2,3\cdots n)$$ here, $H_{i}$ represents the entropy of the jth indicator, and k=1/lnn guarantees $0\leq H_{i}\leq 1$.

(4) Calculate the weights of each indicator
(8)$$W_{i}=1-H_{i}/\sum_{i=1}^{m}(1-H_{i})$$ here, $W_{i}$ represents the weight coefficient of the jth indicator. The entropy weight method is used to calculate the weights of different index layers, which can effectively prevent error transmission [53]. To maximize the accuracy of index calculation, the improved entropy weight method could be used to calculate the weight of indexes, to distinguish differences and identify independence of indexes during dimensional evaluation and comprehensive evaluation.

#### 2.3.5. Construction of the Evaluation Index System

The suitability level of urban construction land is the result of the interactions of each evaluation factor; therefore, the selection of evaluation factors is essential to ensure the accuracy of results. According to the conceptual framework for SEUCL mentioned above and previous research findings of urban land evaluation [3,16], a SUCL evaluation index system is established. The index system is divided into four grades: The indexes of the first grade are the evaluation index of SUCL, which comprehensively indicate the overall level of urban construction land suitability in the upper reaches of Yangtze River. The second evaluate the suitability of the urban construction land from the four different dimensions of the resources carrying capacity (RCC), eco-environmental constraints (EEC), social and economic coping capacity (SECC), and urban developmental potential (UDP). RCC and EEC correspond to carrying capacity, while SECC and UDP correspond to construction suitability evaluation [31]. The third consist of nine support indexes: Land resource carrying capacity, water resource carrying capacity, ecological fragility, eco-environmental sensitivity, population agglomeration, economic development, economic development potential, urbanization level, and transportation superiority [51]. The fourth include 21 supporting indexes. Integrating the characteristics of the urban development of the upper reaches of the Yangtze River and the special requirements for suitability evaluation, four groups of indexes (comprising nine separate geo-environmental subfactors) were constructed for the SEUCL in the upper reaches of Yangtze River.

Level-1 indexes include carrying capacity and urban functions within the conceptual framework for SEUCL. Among level-2 indexes, RCC reflects the maximum carrying capacity of regional resources to afford urban construction or human activities. For example, supporting capacity of land resources and water resources to human activities or urban construction. The indexes include the per capita cultivated area and water resources per capita [46,52]. Eco-environmental constraints are used to measure constraints the ecological environment imposes on urban construction. Attention was focused on ecological vulnerability and ecological sensitivity, covering hazard vulnerability, geological hazard risk, geological hazard risk, topographic relief, and vegetation coverage. Urban land functions are composed of SECC and UDP [13]. Specifically, SECC is used to reflect the ability and level of the existing population and the economic basis of a region to support the future urban development scale [46]. SECC is composed of population agglomeration, economic development, and development potential. These can be further divided into population density, growth rate of the population, GDP per capita, GDP per land, proportion of secondary and tertiary industries, per capita consumption expenditure, per capita fixed asset investment, per capita public budget expenditure [47]. The UDP reflects the potentials of future urban development or possible construction scale by social economic factors of the region. These factors include infrastructure, industrial structure, urbanization rate, and traffic condition [43]. It mainly covers urbanization level and transportation superiority, specifically including urban land use intensity, proportion of urban land area, proportion of urban population, traffic density, trunk road density, and location advantage [32]. The weight in the evaluation of each dimensions are calculated, and the index weight of comprehensive evaluation is the weight used for the comprehensive evaluation in Table 1.

## 3. Results

### 3.1. The Spatial Pattern of SUCL

Single factor evaluation was used to calculate the actual values of the four layers. Then, the weighted sum method was used to evaluate the factors of each dimension. The comprehensive evaluation value of SUCL was calculated, and the spatial variation law of SUCL was identified. Finally, the results of SEUCL and each evaluate dimensions were divided into five grades (I, II, III, IV, V) by applying a natural breaking point approach, corresponding to the following five area classifications: Priority, suitable, sub-suitable, restricted, and prohibited construction areas, as shown in Figure 3.

#### 3.1.1. Spatial Pattern of RCC

RCC is a comprehensive reflection of both regional cultivated land resources and water resources, which can provide basic support and guarantee for urban construction. The value ranges between 0.11 and 0.90, the minimum was found in Chengduo County, which is part of the Yushu Tibetan Autonomous Prefecture, Qinghai Province, and the maximum was found in Muchuan County, Sichuan Province. The land area with high value regions (grades IV to V) of RCC accounts for 26.33%, and the proportion of their resident population is 17.29%. The spatial pattern of the RCC was divided into four quadrants with Chengdu at the center (Figure 3a). The regions of the first quadrant are mainly distributed in the upper reaches of the Jialing River, where the population density is lower compared with the Chengdu Plain; however, the per capita cultivated land resources and water resources are relatively rich. The regions of the second quadrant are mainly distributed throughout the eastern edge of the Tibetan Plateau, where the terrain is relatively flat and the per capita cultivated land resource is relatively rich. For example, Songpan County occupies the 39th place among 346 counties, with a per capita cultivated land area of 0.27 ha. The regions of the third quadrant are mainly distributed in river valleys south of Sichuan Province, where the values of land and water resources distribution are high. The regions of the fourth quadrant are distributed along the middle and lower reaches of the Wujiang River, where high-value regions of water resources are distributed. In addition, the high value areas of RCC are also distributed intermittently along the upper reaches of the Yangtze River valley. Low-value areas of RCC are mainly distributed in the headwaters of the Yangtze River, such as the Qinghai–Tibet Plateau and the middle and upper reaches of the Jinsha River. These areas are characterized by high mountains and deep valleys, interlaced with mountains and rivers, and a scarcity of cultivated land resources.

#### 3.1.2. Spatial Pattern of EEC

Since all evaluation indexes of the ECC are negative indexes, to facilitate the comprehensive evaluation and calculation of the SUCL, the ECC is calculated as reverse. Therefore, the value of ECC is actually a reverse result, i.e., the greater its value, the more suitable it is for urban construction. The high value area of ECC (grades IV to V) spans a total of 97 counties (34 grade V counties, and 63 grade IV counties). The land area accounts for 31.47%, the proportions of resident population are 62.59%, and the average altitude is 2096.46 m. The ECC is strong in the upper reaches of the Yangtze River. The high EEC values are concentrated in the Sichuan Basin. The Yunnan-Guizhou plateau is the main distribution area for ECC grades III and IV, because their elevation, topographic relief, and hazards are lower than the Tibetan Plateau. The low value of RCC (grade I) is mainly distributed in three areas. One such area is the plateau and mountainous area west of Sichuan Basin, which mainly includes Aba Tibetan and Qiang Autonomous Prefecture as well as Ganzi Tibetan Autonomous Prefecture of Sichuan Province, e.g., Li County, Xiaojin County, and Heishui County. The values are below 0.62, which represents the lowest level. The second area is the transition zone from the Tibetan Plateau to the Sichuan Basin, and includes Mianzhu County (0.56), Maoxian County (0.56), Wenchuan County (0.46), and Dujiangyan City (0.47) among others. The third area is the section of the Yangtze River trunk stream from Chongqing to Yichang, especially the Three Gorges Reservoir area, including Yunyang, Fengjie, Wushan, and Badong counties (Figure 3b).

#### 3.1.3. Spatial Pattern of SECC

A total of 24 county-level administrative units were identified in the high value area of SECC (grade V), accounting for 0.39% of the land area, but 23.14% of the GDP. SEEC grade IV contains 87 county-level administrative units, accounting for 13.11% of the land area, and 44.35% of the GDP. The GDP of the 111 county-level administrative units in the high values of SECC accounts for about 70% of the total GDP of the region. As shown in Figure 3c, the high-value regions (grade V) re concentrated in the downtown areas of each city. The evaluation values of downtown Chengdu and the main urban area of Chongqing all exceed 0.8, and the Yuzhong District is 0.95, which is the highest value. There are 24 administrative units (grade V) with seven seats in Chengdu and six seats in Chongqing. The second-highest value region of SECC (grade IV) mainly extends outward from the four provincial capitals, and is concentrated in the Chengyu urban agglomeration and the Yangtze River trunk Yibin to Chongqing section. This area includes Ziyang City, Neijiang City, Yibin City, Zigong City, and Luzhou City in Sichuan Province. The average SECC value of this region is 0.52. Low value regions (grade I to II) are mainly distributed in the Tibetan Plateau, the Sichuan Basin, and its surrounding regions.

#### 3.1.4. Spatial Pattern of UDP

The spatial pattern of UDP is similar to that of SECC, but with even higher concentrations and more significant regional differences. High-value regions (grades IV to V) contain 86 county-level administrative areas, including 50 with grade V, and 36 with grade IV, the land areas of which account for 21.56% and 5.57%, respectively, and the resident populations account for 24.32% and 12.76%, respectively. The low-value area (grade I to II) is larger than the high-value area, and grade I and grade II land areas account for 36.09% and 40.35%, respectively. Chengdu Plain and the counties (districts) around Chongqing are the most concentrated areas with high UDP values, followed by Kunming and Guiyang. The maximum value of UDP was 0.73 in the Qingyang District, Sichuan Province, which formed a high value zone between Chengdu City and Mianyang City. Low-value areas of UDP were concentrated in the transition zone from Sichuan Basin to the Tibetan Plateau and the mountainous areas around the basin. This area includes the Qin-ba Mountain in the north, the Guizhou Plateau in the south, the Western Sichuan Plateau in the west, the Wushan Region in the east, and the Qinghai–Tibet Plateau in the northwest (Figure 3d).

### 3.2. Comprehensive Evaluation and Spatial Pattern of SUCL

The values of SUCL in the upper reaches of the Yangtze River range between 0.00 and 0.76. The maximum value is 0.117, located in Shiqu County, Sichuan Province, which is the westernmost part of Sichuan Province and belongs to the Qinghai–Tibet Plateau region. In Qingyang District, Chengdu City, Sichuan Province, which belongs to the Chengdu Plain, the minimum value was 0.76. The priority and suitable construction area consist of 68 county-level administrative units, the land area of which accounts for only 4.59%, but 49.69% of GDP, and 31.39% of the population. The land of restricted and prohibited construction area accounts for 79.10% (Table 2). This indicates that the SUCL in the upper reaches of the Yangtze River is at a low level and the urban construction should be mainly based on ecological protection.

In terms of the spatial pattern, the spatial differentiation of SUCL is obvious, showing the characteristics of high in the middle and low in the vicinity (Figure 4). For example, the average in Chengdu, Sichuan Province, was 0.46, while the average in western Ganzi Tibetan Autonomous Prefecture was 0.24. The high value area of SUCL is mainly distributed in four regions in the upper reaches of the Yangtze River. Firstly, Chengdu Plain, which is mainly distributed along the economic belt and the major traffic trunk line. Specifically, Mianyang City (with a mean value of 0.41), Meishan City (with a mean value of 0.38), Deyang City (with a mean value of 0.43), and other areas located in the Chengdu Plain are concentrated distribution areas with high values. Secondly, priority and suitable construction areas are mainly distributed in urban agglomerations in the south of Sichuan Province, such as Yibin, Zigong, Neijiang, and other cities. The evaluation value of Cuiping District of Yibin City is 0.56, which is the highest value in this region. Thirdly, Chongqing Urban City, along the Cheng-Yu freeway, and the Chongqing to Yibin section of the Yangtze River trunk stream, form a high-value SUCL region, which includes the Chengdu-Chongqing railway along the Ziyang City, Neijiang City, Yongchuan District, and Rongchang District. Fourthly, Guiyang and Kunming, the two provincial capitals at the core, form a high-value cluster area of SUCL in the shape of circles. The highest value of Guiyang is located in Nanming District (0.55) and the highest value of Kunming is located in Guandu District (0.46).

### 3.3. The Spatial Distribution Mechanism of SUCL

As shown in Figure 5, the SUCL hot spots value in the upper reaches of the Yangtze River display obvious spatial heterogeneity. Significant regional differences exist in the distribution of the RCC values across the upper reaches of the Yangtze River. Hot spots with 99% confidence level are mainly distributed in the transition area from Sichuan Basin to the western Sichuan Plateau and the mountainous areas along the eastern edge of the study area (Figure 5a). The cold spots with 99% confidence level are mainly distributed in the following three regions: The source of the Yangtze River, the surrounding area of Kunming, and the Wumeng mountain area. Non-significant areas are mainly distributed in the Sichuan Basin, Western Sichuan, and Yunnan Province. In contrast to RCC, the hot spot and cold spot distributions of EEC have opposite characteristics, and form complementary characteristics with RCC in the spatial domain (Figure 5b). For example, 99% of the hot spots with confidence level are mainly distributed in the Sichuan Basin, where RCC is not significant. The EEC cold spot areas are distributed in the headwaters of the Yangtze River, the eastern part of the Tibetan Plateau, and the Three Gorges reservoir area. Among these, the eastern edge of the Tibetan Plateau is the most concentrated, while the RCC is not significant in this area. In addition, areas with non-significant EEC are mainly distributed in the surrounding areas of the basin. In contrast, RCC forms a number of hot spots in this area. Such distribution characteristics are related to a mismatch between the spatial distribution of population and resources in the upper reaches of the Yangtze River. The SECC hot spots with a confidence level of 99% are distributed in the Sichuan Basin and around Zunyi City, Guizhou Province. A transition zone with a confidence level of 90–95% formed between these two places (Figure 5c). The cold spots with a confidence level of 90–95% are distributed in the transition zone from the Sichuan Basin to the Tibetan Plateau and the headwaters of the Yangtze River. This distribution feature is similar to EEC, but the clustering area of cold spots and hot spots is smaller, and the spatial distribution of non-significant areas is more distributed. The cold spots and hot spots of UDP in the research area show obvious circular distribution characteristics. The hot spots with 99% confidence level are concentrated in the Sichuan Basin, but their areas are smaller than those of SECC. Hot spots tend to extend to Guizhou Province, but unlike SECC, the value of UDP does not form another hot spot region in Guizhou Province. In the Sichuan Basin, cold spots and non-significant areas form a circle-like structure (Figure 5d). SCUL is a comprehensive reflection of four-dimensional evaluation factors, and the spatial distribution of SUCL cold spots and hot spots is most similar to SECC. Hot spots are concentrated in the Sichuan Basin and the Guizhou Plateau. Cold spots are distributed in Jinsha River above Yibin, and the headwaters of the Yangtze River (Figure 5e).

## 4. Discussion

### 4.1. The Influence of the Physical Geographical Environment

The existing research draws the common conclusion that the SUCL is restricted by the natural geographical environment, which includes the terrain, water sources, climate, and other factors [8,19,54]. The high value areas of SUCL are mainly concentrated in the Sichuan Basin and surrounding central cities, such as Guiyang and Kunming. These regions belong to areas that benefit from a particularly advantageous natural geographical environment, associated with relatively flat terrain, hydrothermal conditions suitable for agricultural production, few geological disasters, and little constraints of the ecological environment. This finding is consistent with the results of Tang et al. [32]. From the perspective of environmental constraint, plateaus and mountain areas are more fragile than basins and river valleys [55]. Hazard vulnerability and risks are higher, because of higher average altitude and greater topographic relief. The reasons are that these regions have enormous topographic relief, and unstable monsoon climate, which leads to heavy summer rains and frequent mountain hazards [56]. Although plateau and mountain areas have more arable land and water resources per capita than basin areas, this advantage is insufficient to offset the other three disadvantages. Therefore, the SEUCL in mountainous areas needs to fully consider the influencing factors of mountain disasters to avoid the negative effects of these disasters on urban construction. Consequently, this area should focus on controlling the disorderly expansion of construction land, and the protection of the ecological land.

In terms of SECC, superior physical geographical conditions in the Sichuan Basin for urban construction provided a good foundation for agricultural development, and consequently, Chengdu and Chongqing became two large cities. In the process of their development, the central city forms a siphonic effect, which yields a dense population and an active economy for the region [57,58]. The per land GDP and per capita GDP of the 96 counties in the Sichuan Basin are 12.35 times and 4.26 times higher than that of other regions, respectively. A large number of small and medium-sized towns as well as urban agglomerations have formed around the two central cities and along the traffic arteries that connect these central cities, such as the Chengyu Urban Agglomeration [59] and the southern Sichuan Urban Agglomeration, which provide suitable social and economic conditions for urban construction.

The UDP is closely related to factors such as the level of urbanization and traffic accessibility [16,60]. The basin region has a more active economy, higher per capita consumption level, and fixed asset investment level. In addition, the relatively flat terrain simplifies infrastructure construction, which is conducive to the formation of a dense transportation network and trunk lines, and also promotes the central city to have closer economic ties with other towns, thus increasing the development and construction potential of cities and towns. Therefore, more effective measures should be implemented in the future development to overcome the limitations induced by the natural geographical environment. Furthermore, the construction of infrastructure should be strengthened, and the development of cities and towns with better natural environment should be prioritized.

### 4.2. The Pole–Axis Effect Affects the SUCL

The pole–axis system is one of the main models of the Chinese urban spatial structure. The poles refer to central cities within the region, while the axis is the main traffic arteries connecting the poles, such as railways, highways, and rivers [61,62]. The SUCL in the upper reaches of the Yangtze River shows the spatial structure characteristics of pole–axis distribution. Such a spatial distribution pattern forms by “point” and “axial” drivers in the region. These cause agglomeration of nearby population, resources, and other elements, thus forming a higher SECC and UDP.

On the one hand, the radiation effect of the pole is obvious. The four central cities in the upper reaches of the Yangtze River are core poles for regional development. As shown in Figure 5b–d, EEC, SEE, and UDP all use the four central cities as core poles to form high-value areas, which indicates that these central cities have a strong driving ability for the construction of towns in surrounding areas. This phenomenon can be attributed to superior infrastructures, political, and cultural functions of central cities. These central cities therefore attract population and industrial agglomeration and have stronger advantages with regard to the population density and growth rate of the population [63]. Population agglomeration can produce higher GDP and increase household expenses and government public expenditures, thus driving the growth of SECC in surrounding regions. In addition, because of the needs for urban construction, central cities tend to have more public expenditure, especially in transportation construction, to strengthen the connection with other central cities and to increase transportation superiority [64]. Urban land use intensity, proportion of urban land area, and proportion of urban population will increase significantly. Numerical values of traffic density, trunk road density, and location advantage are higher, and the numerical value of transportation superiority increased further, thus equipping both the central cities and their surrounding regions with higher UDP. Therefore, the future urban construction should continue to expand outward around the vital poles, or alternatively, new poles should be formed to drive regional urban construction.

On the other hand, the areas distributed along the axes including high-speed railways, expressways, and the Yangtze River, have stronger SUCL. A concentrated distribution area with high values of SUCL has formed along the Cheng-Mian-Le intercity high-speed railway (Figure 6a), the Chengyu high-speed railway or expressway (Figure 6b), the Yugui high-speed railway (Figure 6c), and the Yangtze River trunk Yibin to Chongqing section (Figure 6e). It is worth noting that cities along the Yangtze River have higher carrying capacity of land resources and water resources, and correspondingly, their value of comprehensive SCUL is also higher. Traffic road construction is an effective way for the government to both improve regional infrastructure and promote regional economic development. Firstly, regions along the arterial traffic, especially highway and high-speed railways, can agglomerate frequently business activities [65]. These areas, with their higher transportation superiority, have become population and economy aggregation areas, featuring a higher population concentration and economic development level than plateau and mountainous areas. Secondly, arterial traffic often connects central cities with other cities, which can bring capital, population, and technological elements of central cities to cities along the traffic lines [66,67]. Hence, cities surrounding the central axis have higher social and economic coping capacity. Rivers open to navigation also exert the same functions. For example, the Yibin-Chongqing Section of the Yangtze River is called the “gold watercourse” that connects Yibin, Luzhou, Hejiang, and Jiangjin. These cities receive the higher evaluation values in SECC and UDP, thus showing higher SUCL.

In the future, the government should invest more in the construction of such axes to further benefit from their driving function. Alternatively, the government should build new important axis, especially in areas with sub-suitable and restricted resources, to improve the SCUL.

### 4.3. Advantages and Limitations

Although some publications have evaluated SUCL, few established an effective relationship between land function and urban construction. This study offers three major advantages: First, an organic connection between land resources and human activities was established for the SEUCL. Consequently, the framework in the present study was proposed to reacquaint the carrying capacity of the land ecosystem and the suitability of the land, to realize goal-oriented and holistic evaluation of SUCL. Second, with regard to the application methods, this study designed a new method for the suitability evaluation of urban construction land based on four dimensions (RCC, EEC, SECC, and UDP). The approach of using administrative regions as evaluation units was also improved, by using the upper reaches of the Yangtze River Basin as a whole for evaluation, and adding both the disaster vulnerability and disaster risk assessment indexes. Thus, the calculation method of SEUCL could be enriched. Third, this study analyzed the suitability of SUCL in the upper reaches of the Yangtze River from the whole and from four different dimensions, which identified the spatial distribution characteristics of their suitability through hot spot analysis (Getis-Ord Gi*). It has become clear that the high value areas of SUCL are mainly concentrated in the Sichuan Basin and surrounding central cities, because of their advantageous natural geographical environment and the pole–axis effect. In addition, the entropy weight method was improved to ensure the integrity of data and to reduce the impact of extreme data points on the evaluation results.

This study attempted to establish an evaluation index system suitable for the ecologically sensitive areas of the upper reaches of the Yangtze River. A series of quantitative indicators was established, which was closely related to SUCL based on the unit at the county level. However, it must be acknowledged that there are still a number of points that need to be considered in future studies. Firstly, governmental policies, cultivated land protection degree, biological resources protection degree, and other important indicators related to urban construction land were not considered. Secondly, different indexes have their own means of collection, and the original data were acquired from multi-scales, which inevitably generates uncertainties in the spatial analysis. More reliable evaluation indexes should be adopted to improve the evaluation.

Urban expansion is closely related with land ecological health [64,68]. For example, epitaxial extensive applications cause problems such as construction land expansion, traffic jams, and ecological damage, thus threatening ecological health of lands in many regions, especially in ecologically sensitive regions [69]. In the SEUCL of the upper reaches of the Yangtze River, ecological health of lands is used as a constraint of intensive land use, which can realize the double effect of land intensive use and ecological health. According to the suitability evaluation, a large area is restrained by ecological environment in the upper reaches of the Yangtze River (the land area of EEC grades I to III accounts for 68.53%). This is controlled by landform conditions, geological disasters, and vegetation coverage. In the future, attention should focus on the control of the urban expansion in regions with vulnerable ecology or that are sensitive to ecological changes. The most appropriate urban construction region can be identified through suitability evaluation, thus realizing a sustainable and intensive land use in cities.

## 5. Conclusions

Based on four dimensions (RCC, EEC, SECC, and UDP), using multi-factor comprehensive evaluation model, improved entropy method, and GIS analysis method, this study evaluated SUCL of the upper reaches of the Yangtze River. The proposed method considers the actual situation of ecologically sensitive areas, evaluates the suitability of four dimensions, and identifies their spatial distribution characteristics and influence mechanisms. The results show that the value of SUCL in the upper reaches of the Yangtze River ranges between 0.00 and 0.76, and priority and suitable construction areas account for only 4.59%; however, the area of restricted and prohibited construction area accounts for 79.10%. Generally, the SUCL in the upper reaches of the Yangtze River has a low level and urban construction should be mainly based on ecological protection. The high value area of SUCL is mainly distributed in the Chengdu Plain, while urban agglomerates are mainly distributed in the south of Sichuan Province, Chongqing Urban City, Guiyang City, and Kunming City (in the shape of circles). Hot spots are concentrated in the Sichuan Basin and Guizhou Plateau, while cold spots are distributed in Jinsha River above Yibin, and the headwaters of the Yangtze River. These distribution characteristics are influenced by both the physical geographical environment and the pole–axis effect. In the future, the government should increase investment in the construction of similar axes to further utilize their driving role, and establish new important axes, especially in sub-suitable and restricted areas, to overcome the natural preconditions of SCUL restrictions. Furthermore, the development strategy should be adjusted to local conditions, and different suitability grades should be considered to establish short-term or long-term urban construction plans.

In spite of the associated limitations and uncertainties, the results highlight the advantages of this method for the suitability evaluation of urban construction land based on four dimensions (RCC, EEC, SECC, and UDP). Moreover, this approach also provides a new perspective for studying the spatial distribution of SUCL. Valuable information is needed for policy makers to implement urban land management in the upper reaches of the Yangtze River. The findings of this study provide reliable information for the effective formulation of urban construction plans, while also providing a solid basis for determining urban construction and ecological protection targets during regional planning.

## Figures and Tables

**Figure 1 ijerph-18-04252-f001:**
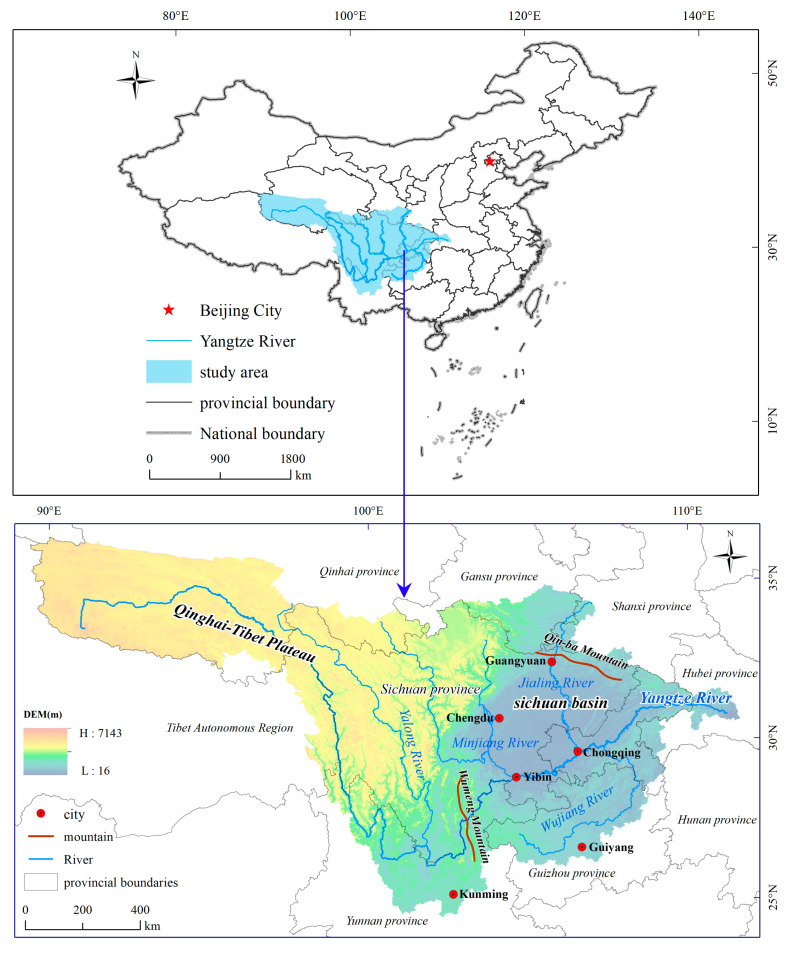
Location of the study area.

**Figure 2 ijerph-18-04252-f002:**
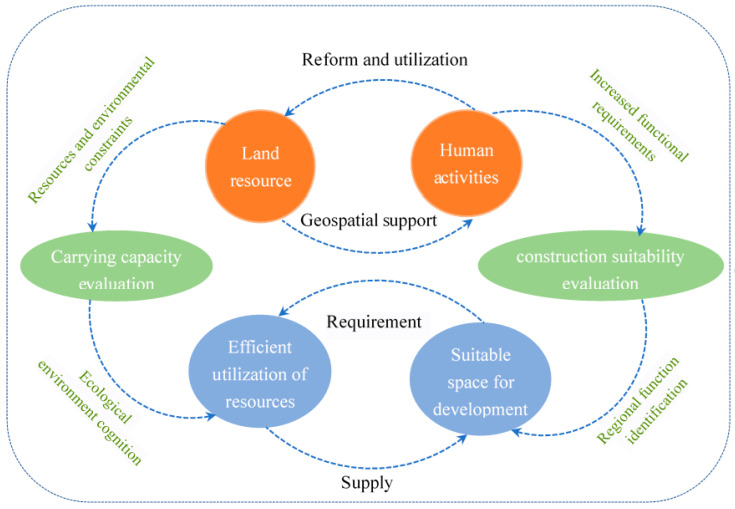
Theoretical logic of suitability evaluation of urban construction land (SEUCL).

**Figure 3 ijerph-18-04252-f003:**
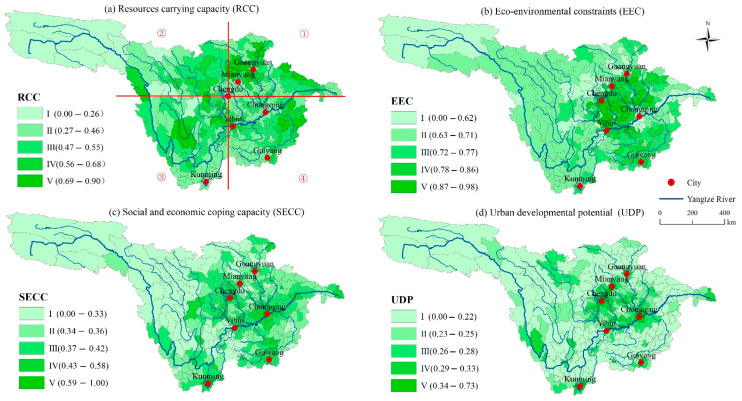
Suitability of urban construction land (SUCL) at different dimensions.

**Figure 4 ijerph-18-04252-f004:**
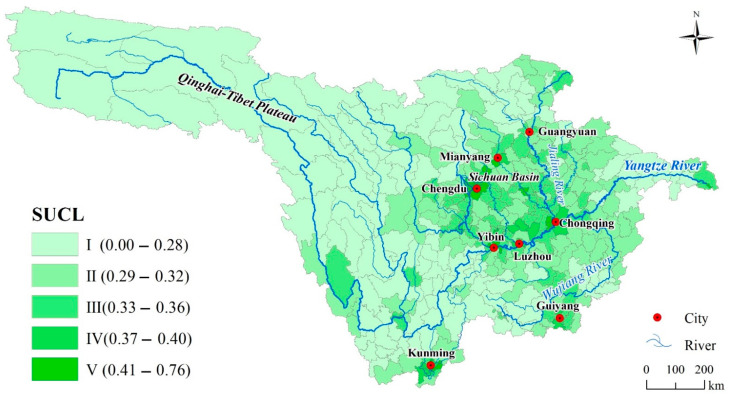
Results of comprehensive evaluation and spatial pattern of suitability of urban construction land (SUCL).

**Figure 5 ijerph-18-04252-f005:**
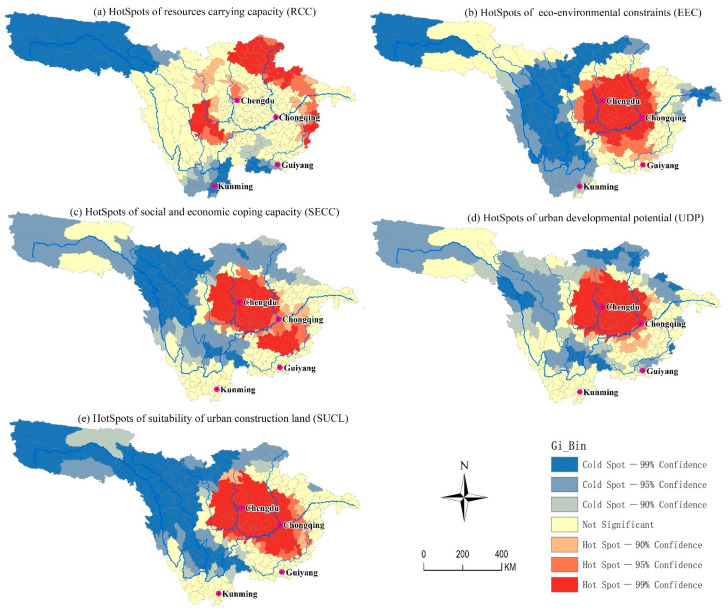
Spatial distribution of hot spots and cold spots of suitability of urban construction land SUCL.

**Figure 6 ijerph-18-04252-f006:**
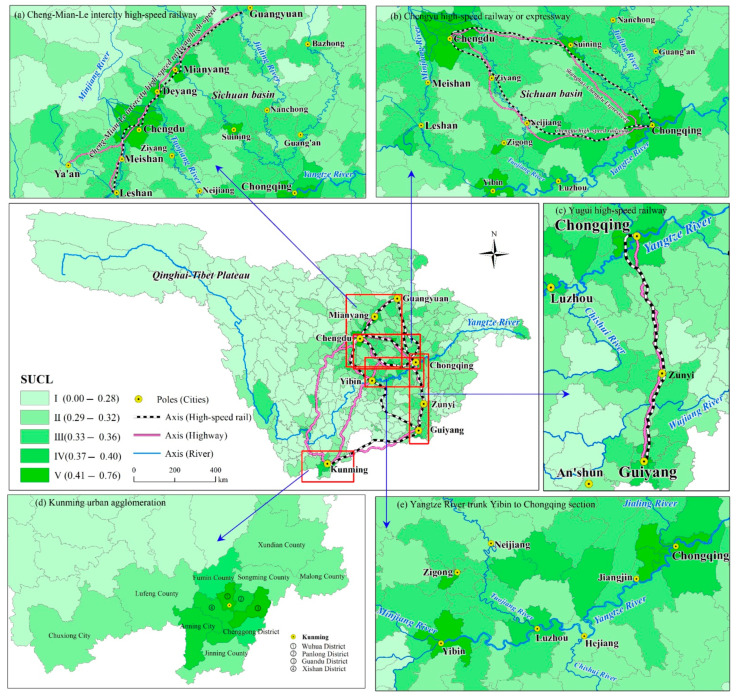
Pole–axis effect of the suitability of urban construction land (SUCL) spatial distribution.

**Table 1 ijerph-18-04252-t001:** Index system and index weights of the suitability evaluation of urban construction land (SEUCL).

First Grade Index	Second Grade Index	Third Grade Index	Fourth Grade Index	Index Weight of Different Dimensions Evaluation	Index Weight of Comprehensive Evaluation	Index Properties
Suitability of urban construction land (SUCL)	Resources carrying capacity (0.143)	Land resource carrying capacity	Per capita cultivated area	0.432	0.062	+
		Water resource carrying capacity	Water resources per capita	0.568	0.042	+
	Eco-environmental constraints (0.237)	Ecological fragility	Hazard vulnerability	0.135	0.032	−
			Geological hazard risk	0.233	0.038	−
		Eco-environmental sensitivity	Average elevation	0.281	0.026	−
			Topographic relief	0.192	0.051	−
			Vegetation coverage	0.159	0.031	+
	Social and economic coping capacity (0.363)	Population agglomeration	Population density	0.107	0.071	+
			Growth rate of the population	0.091	0.022	+
		Economic development	GDP per capita	0.201	0.065	+
			GDP per land	0.082	0.083	+
			Proportion of secondary and tertiary industries	0.124	0.034	+
		Development potential	Per capita consumption expenditure	0.206	0.031	+
			Per capita fixed asset investment	0.068	0.033	+
			Per capita public budget expenditure	0.121	0.046	+
	Urban developmental potential (0.257)	Urbanization level	Urban land use intensity	0.249	0.056	+
			Proportion of urban land area	0.132	0.049	+
			Proportion of urban Population	0.137	0.052	+
		Transportation superiority	Traffic density	0.152	0.071	+
			Trunk road density	0.129	0.052	+
			Location advantage	0.201	0.053	−

Note: “+” represents positive indexes, and “−” represents negative indexes.

**Table 2 ijerph-18-04252-t002:** Classification of suitability of urban construction land (SUCL) in the upper reaches of the Yangtze River.

Comprehensive Classification and Value	Suitable Type of Construction	Number of Administrative Regions	Proportion of Land Area (%)	Proportion of Population (%)	Proportion of GDP (%)
Ⅰ (0.00–0.28)	Prohibited	72	50.66%	10.23%	4.29%
Ⅱ (0.29–0.33)	Restricted	109	28.61%	27.09%	17.64%
III (0.34–0.37)	Sub-suitable	97	16.14%	31.29%	28.38%
Ⅳ (0.38–0.41)	Suitable	30	3.12%	12.36%	14.55%
Ⅴ (0.42–0.77)	Priority	38	1.47%	19.03%	35.14%

## Data Availability

The data presented in this study are available on request from the corresponding author.

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
