# Peer review of "How to Identify Future Priority Areas for Urban Development: An Approach of Urban Construction Land Suitability in Ecological Sensitive Areas"

_ijerph, 2021, doi:10.3390/ijerph18084252_

Round 1
Reviewer 1 Report
I appreciate the authors’ effort to view urban construction land (UCL) as a coupled human-nature (or social-ecological, or - environmental) system and evaluate its critical suitability comprehensively and reasonably. The outcomes of the findings, as exemplified in the case of this study in question, are essential to guide the decision-making for urban land use and development. I would recommend minor revisions at which detailed below:
- The index selection seems quite vague, although the authors claimed that “the SUCL evaluation index system is established based on adaptive, scientific, sustainable, representative, and operability principles” (L267-8). Is there any existing framework or established research that guides the selection of these indexes? Please add clarifications for this.
- L188-204, it is not clear about the “decomposition” of SUCL. To be specific, the authors failed to clarify whether (1) and (2) are two components of the conceptual framework or two sequential procedures of the methodology. Otherwise, it does not make sense how SUCL could be a “part” of SUCL. Some statements in this paragraph may need rephrasing as they may confuse the readers, i.e., “The forms the suitability of the land for a certain type of human activity or construction, i.e., suitability” (L201-202).
- The authors used an improved entropy method to calculate the SEUCL index weight and applied them to the fourth- (lowest-tier) indexes. Considering the index hierarchy, readers may ask whether the indexes of upper tiers have also been weighed or just treated equally when calculating the comprehensive evaluation of SUCL in Section 3.2? If not, why?
- Section 4.2 (start from Line 489): the causal mechanisms between the pole-axis effect and the SUCL need to be justified. Is that a unilateral relationship in which the pole-axis effect can affect the SUCL, or there is causal reciprocity between the two—suggesting that SUCL also be a factor that can generate the pole-axis effect?
- I would also suggest the improvement in mapping, Figure 1 in particular. There have been plenty of places identified in this manuscript whose corresponding titles are hugely absent on the maps. For some cardinal and most frequently used ones, i.e., the major rivers and provincial/autonomous regions, Qinghai-Tibet Plateau (or Tibetan Plateau), and Sichuan Basin, the authors would need to label them on the maps; otherwise, readers may get lost when reading the text about the maps.
- Some minor errors: 1) SEUCL instead of SUECL (L125); 2) minimum or maximum? (Line 379); 3) “The” approach not “To” approach (L533); 4) L301-2: Chengdu County (or Chindu), based on my knowledge, is under the justification of Yushu Tibetan Autonomous Prefecture, Qinghai Province, not Tibet Autonomous Region, please double check though. Make sure to remove all mechanic errors before resubmission.
Reviewer 2 Report
Dear Authors,
Thank you for the manuscript with the title "How to Identify the Future Priority Areas for Urban Development: An Approach of Urban Construction Land Suitability in Ecological Sensitive Areas". Several comments regarding the presented manuscript are as follow:
- The used terms presented in the manuscript for example "urban construction land" must be corrected in to "Urban land";
- The Ecological aspects from selected regions with a focus on health can be analysed in the analysis results will be very interesting.
The manuscript with the title "How to Identify the Future Priority Areas for Urban Development: An Approach of Urban Construction Land Suitability in Ecological Sensitive Areas" does not meet the aims and scope of the journal "International Journal of Environmental Research and Public Health" to which the article is submitted. The manuscript with a presented content can be sent to the other journal (for example: "Land use Policy").
Reviewer
Reviewer 3 Report
This paper reports a research concerning the relevant issue of suitability.
I red the paper that responds to the publishing required standard. Therefore it can be published.
Author Response
Thank you very much for the positive evaluation of the reviewer. We have checked and corrected the full text again to make it more in line with the requirements of the journal.